



# Precipitation regimes over central Greenland inferred from 5 years of ICECAPS observations

Claire Pettersen[1], Ralf Bennartz[1,2], Aronne J. Merrelli[1], Matthew D. Shupe[3,4], David D. Turner[4] and Von P. Walden[5]

[1]Space Science and Engineering Center, University of Wisconsin – Madison, Madison, Wisconsin, USA
[2]Vanderbilt University, Nashville, Tennessee, USA
[3]Cooperative Institute for Research in Environmental Science, University of Colorado
[4]NOAA – Earth System Research Laboratory, Boulder, Colorado, USA
[5]Washington State University, Pullman, Washington, USA

*Correspondence to*: Claire Pettersen (claire.pettersen@ssec.wisc.edu)

**Abstract.** A novel method for classifying Arctic precipitation using ground based remote sensors is presented. Using differences in the spectral variation of microwave absorption and scattering properties
of cloud liquid water and ice, this method can distinguish between different types of snowfall events depending on the presence or absence of condensed liquid water in the clouds that generate the precipitation. The classification reveals two distinct, primary regimes of precipitation over the Greenland Ice Sheet (GIS): one originating from fully glaciated ice clouds and the other from mixed-phase clouds. Five years of co-located, multi-instrument data from the Integrated Characterization of
Energy, Clouds, Atmospheric state, and Precipitation at Summit (ICECAPS) are used to examine cloud and meteorological properties and patterns associated with each precipitation regime. The occurrence and accumulation of the precipitation regimes are identified and quantified. Cloud and precipitation observations from additional ICECAPS instruments illustrate distinct characteristics for each regime. Additionally, reanalysis products and back-trajectory analysis show different synoptic-scale forcings
associated with each regime. Precipitation over the central GIS exhibits unique microphysical characteristics due to the high surface elevations as well as connections to specific large-scale flow patterns. Snowfall originating from the ice clouds is coupled to deep, frontal cloud systems advecting up and over the southeast Greenland coast to the central GIS. These events appear to be associated with individual storm systems generated by low pressure over Baffin Bay and Greenland lee cyclogenesis.





Snowfall originating from mixed-phase clouds is shallower and has characteristics typical of supercooled cloud liquid water layers, and slowly propagates from the south and southwest Greenland along a quiescent flow above the GIS.

## 1 Introduction

The Greenland Ice Sheet (GIS) is losing mass at an accelerating rate (Tedesco et al., 2011). Snowfall is the primary source of mass of the GIS, while precipitation variability is the main driver of inter-annual variability in the mass balance of the GIS (van den Broeke et al., 2009). Airmass origins and mechanisms that result in precipitation over the central GIS are largely unknown, therefore estimates of snowfall accumulation over the GIS are not well constrained (Thomas et al., 2000). Climatological ice

core studies of the GIS show that precipitation accumulation in central Greenland has historically been most affected by changes in atmospheric dynamics as opposed to temperature (Kapsner et al., 1995). Additionally, studies have shown that individual storm systems are the major driver for snow accumulation in the central GIS (Bromwich et al., 1998; Rogers et al., 2004; Schuenemann et al., 2009). We can further test these claims by connecting remote sensing and in-situ observations of the

atmosphere during precipitation events over the GIS to the accompanying large-scale dynamics.

     Here we use a combination of remote sensing and in-situ measurements of snowfall over Summit Station, Greenland, to study precipitation characteristics over the GIS. We further use reanalysis and back-trajectory analysis to put different precipitation regimes into a synoptic-scale context. Our analysis relies primarily on observations made by the Integrated Characterization of

Energy, Clouds, Atmospheric state, and Precipitation at Summit (ICECAPS; Shupe et al., 2013) suite of instruments. The key ICECAPS instruments are the microwave radiometers (MWRs), which we use to separate snowfall into two distinct categories. MWRs make passive measurements of radiance at discrete microwave frequencies and are commonly used both as ground-based and space-borne systems. As shown in previous studies (Spencer et al., 1989; Kneifel et al., 2010; Pettersen et al., 2016), MWR

high frequency (HF) window channels (90, 150, and 225 GHz) are sensitive to ice hydrometeors occurring during snowfall events, whereas the low frequency (LF) window channel (31.40 GHz) is only very weakly affected by ice hydrometeors (Johnson et al., 2012). In this work, we utilize the




differences between LF and HF channels when ice is present, coupled with the ability of the MWR to detect cloud liquid water (CLW), to classify snowfall events into categories: events with snow originating from fully glaciated ice clouds (IC snow), events occurring with a measureable amount of cloud liquid water in the column (CLW snow), and events where we cannot assign a distinct cloud type

(Indeterminate snow). This MWR snow categorization method is illustrated in detail in Sect. 3.

In Sect. 4, we show that the majority of precipitation events, both by occurrence and accumulation, at Summit fall distinctly into the two snowfall regimes defined above: IC snow and CLW snow. The CLW snow events are those with associated classic single- or multi-layer, Arctic mixed-phase clouds. In contrast, the IC snow events are coupled with deep, nimbostratus-like clouds. Arctic

mixed-phase clouds are comprised of both ice crystals and supercooled CLW droplets and are commonly observed throughout the Arctic region and during all seasons (Verlinde et al., 2007; Shupe et al., 2008). Arctic mixed-phase clouds are often long lived and can contribute to accumulation over the central GIS (Morrison et al., 2012; Shupe et al., 2013). Precipitation from the IC events forms by pure ice growth processes within these deep, fully-glaciated ice clouds, which produce pristine and irregular

ice crystals (Korolev et al., 1999; Pruppacher and Klett, 2012).

By applying our MWR precipitation classification method to ICECAPS instrumentation, we observe distinct characteristics that further our understanding of the precipitation processes at Summit Station. Combining the regime classification with available surface meteorological data and reanalysis products, we can infer the dynamics that govern these air masses and their associated regional impacts

by how these events are propagated up and over the central GIS. We will investigate if IC and CLW snow display different vertical profiles as observed by active instruments at Summit Station. These profiles are correlated with features of fully glaciated and mixed-phase clouds, respectively (Sect. 4). Using reanalysis and back-trajectory data, we further study the relationship of IC and CLW events with large-scale forcings (Sect. 5).

**2 Datasets and Methods**

This work employs ground-based remotely-sensed measurements, meteorological observations, qualitative in-situ information, radiative transfer models, and reanalysis products. Though the



measurements are from one point on the GIS, we can use these measurements to inform us about processes occurring over the larger region of the central GIS by connecting to their associated dynamical processes through the reanalyses. In this section, we outline the data and products we employ in the study.

## 2.1 ICECAPS

Summit Station is located at 72° 36′ North, 38° 25′ West at 3,216 meters above sea level, ideally situated to study the atmospheric processes occurring over the central GIS. For almost 30 years, Summit Station has been the site of cryospheric and atmospheric studies, starting with the Greenland Ice Sheet Project 2 (Dansgaard et al., 1993). The U.S National Science Foundation and the National Oceanic and Atmospheric Administration have funded and operated facilities enabling continuous measurements of properties of atmosphere, ice sheet, and snow. The ICECAPS instrument suite has expanded studies of the atmosphere by augmenting Summit Station with a comprehensive remote-sensing and in-situ instrumentation suite (Shupe et al., 2013). One goal of ICECAPS is to better measure the cloud and precipitation processes at Summit and use observations to connect the processes to the energy and mass budgets of the central GIS (Shupe et al., 2013). The ICECAPS suite expands the existing network of Arctic atmospheric observatories (Uttal et al., 2016) and has been operating year-round with dedicated local scientific staff since July 2010. The ICECAPS instrument suite was designed to be similar to successful Department of Energy, Atmospheric Radiation Measurement sites (Ackerman and Stokes, 2003). Here we outline the ICECAPS instruments and retrieval products used in this study.

### 2.1.1 Microwave Radiometers

Observations from passive microwave frequencies are fundamental to this work as they are ideally suited to isolating atmospheric components. The ICECAPS suite has two microwave radiometers (MWRs), each with multiple channels that measure the brightness temperatures (BT) at specific microwave frequencies. The Humidity and Temperature Profiler (HATPRO) has seven channels near the 22.24 GHz water vapour absorption line, and seven channels near an oxygen absorption line from



51-58 GHz (Rose et al., 2005). Additionally, there is a high-frequency MWR (MWRHF; Turner et al. 2009) with two channels: 90 and 150 GHz. The inclusion of the MWRHF increases sensitivity to low liquid water path (Crewell and Löhnert 2003; Turner et al., 2007), and ice (Kneifel, et al 2010), which is needed in the dry Arctic conditions. Here we use the zenith pointing, coincident observations from the

MWRs for downwelling atmospheric radiance measurements in the 31.40 and 150 GHz window channels every four seconds.

We also use retrieved values of the precipitable water vapour (PWV), which uses channels from the HATPRO and MWRHF (MWRRET algorithm; Turner et al., 2007, Cadeddu et al. 2013). MWRRET employs the Monochromatic Radiative Transfer Model (MonoRTMv5.0; Clough, et al.

2005), which includes the more accurate treatment of the width of the 22.2 and 183.3 GHz water vapour lines (Payne et al. 2008) and the water vapour continuum absorption (Turner et al. 2009; Payne et al. 2011). The liquid water absorption model of Turner et al. (2016) is used within MonoRTM for this study; this absorption model provides better accuracy when the cloud liquid is supercooled. However, due to a high bias in the liquid water path (LWP) retrievals during precipitation at Summit, we do not

include MWR-derived LWP values in this study (discussed in Sect. 3; Pettersen et al., 2016).

**2.1.2 Millimetre Cloud Radar**

The ICECAPS suite has a millimetre wavelength cloud radar (MMCR): a zenith pointing, single-polarization, 35 GHz ($K_a$ band; 8 mm wavelength), Doppler pulsed radar (Moran et al. 1998). The MMCR was developed for the Atmospheric Radiation Measurement program to make comprehensive

continuing observations of both clouds and precipitation events at remote locations with minimal maintenance (Kollias et al. 2016). The MMCR has a high sensitivity to the vertical structure of ice and CLW layers. The ICECAPS MMCR product includes profiles of reflectivity, mean Doppler velocity, and Doppler spectral width at a vertical resolution of 45 meters and temporal resolution of 2 seconds.

Hydrometeors with geometric diameters less than 3 mm are in the Rayleigh scattering region for

the MMCR (Kneifel et al., 2011). Precipitation events observed at Summit, qualitatively, fall below this threshold (Castellani et al., 2015). The mean Doppler velocities measure the relative speed of the precipitation falling towards or away from the detector, but they are convolved with any turbulence



present in the vertical winds of the air masses in which the precipitation is embedded. Doppler spectral illustrate the variance of the Doppler velocities within a given pulse volume related to the turbulence, shear, and the spread in particle fall speeds, which relates to the distribution of particle sizes and habits.

Retrieved values of snowfall rate (mm hr$^{-1}$) liquid water equivalent (LWE) were calculated using an empirically derived relationship from Matrosov (2007) defined as:

$$Z_e = 56\ S^{1.2} \qquad\qquad\qquad (1)$$

where $Z_e$ is the maximum reflectivity value found between 200 and 800 meters above the MMCR and S is the snowfall rate in mm hr$^{-1}$ LWE. Though there are differences in the ice habits and distributions for the observed events, this relationship holds well for cases with non-aggregated crystals with negligible amounts of liquid water and riming. Such conditions are often observed at Summit (Matrosov, 2007; Shupe et al., 2013).

### 2.1.3 Precipitation Occurrence Sensor System

The precipitation occurrence sensor system (POSS) is a compact and deployable, continuous wave, X-band Doppler radar (Sheppard and Joe, 2008). The POSS samples approximately one cubic meter of air directly above the transmitter and receiver and is used for surface observations of precipitation type, amount, and frequency. The POSS measures the Doppler velocities and reflectivities of hydrometers. We utilize two products from the processed POSS data: the POSS power units and the retrieved liquid equivalent snow rate. The POSS power unit is simply a value assigned to the zeroth moment of the Doppler spectrum analogous to integrated reflectivity and can be used as a binary indicator of precipitation. The LWE snow rate is based on a precipitation estimation algorithm and associated catch ratio outlined in Sheppard and Joe (2008).

### 2.1.4 Radiosondes

We incorporate data from twice daily balloon-borne radiosondes. The launches have been continuous at Summit Station since July 2010 and occur at approximately 1200 and 2400 Coordinated Universal Time (UTC). The radiosondes are Vaisala models RS-92K and RS-92SGP. The soundings gather in-situ measurements of temperature, pressure, relative humidity, and horizontal wind speed and direction.



### 2.1.5 Ice Particle Imaging Camera

The Ice Particle Imaging Camera (IcePIC) is similar to the snowflake photographing apparatus developed by Libbrecht (2007). During a snowfall event, a scientific technician captures falling ice onto a cold microscope slide (to limit snowflake melt) and then photographs the slide with a Nikon D50

DSLR camera mounted on a ~5.6X magnifying microscope body, which is stored in an outdoor shelter. Though these observations are not quantitative, they are helpful in providing some qualitative evidence as to what ice habits are falling during specific events.

### 2.2 Clear Sky Radiative Transfer

The microwave emission and absorption of the dry gases and the water vapour are modelled using the

radiosonde in-situ measurements of pressure, temperature, and relative humidity. The twice-daily radiosondes are linearly interpolated to the MWR observation times. We then employ MonoRTMv5.0 using inputs of layer temperature, pressure, and relative humidity from the interpolated dataset to compute the clear sky radiance at the MWR observed frequencies. Since this is a clear-sky BT calculation, we do not include any cloud liquid water in the model.

This study compares snowfall events that occur over a span of 5+ years. Consequently there is variance in the MWR BTs that depends on background temperature and water vapour profiles and the seasonal variation. To facilitate comparison of events occurring at different times of the year and with dissimilar atmospheric profiles, we use MonoRTM calculations to account for this variation. We use pressure, temperature, and relative humidity from the interpolated radiosonde data and the resulting

MonoRTM calculations to obtain clear-sky BT values at the HATPRO and HFMWR frequencies. For altitudes above available radiosonde measurements, the U.S. Standard Atmosphere (McClatchy et al., 1972) is used up to 30 km above ground level. We then subtract the calculated clear-sky BTs from observations from the MWR. The resulting ΔBT values are the CLW and/or ice contributions as a function of frequency. Due to the high and dry location of Summit Station, the optical depths of the

atmospheric components at the microwave window channels are very low. Thus, the different contributions to the microwave radiance are approximately additive, and we can employ this method with decent accuracy across the time range of the ICECAPS dataset (further illustrated in Sect. 3).





### 2.3 Reanalysis Data

Section 5 of this work ties the observations of snowfall events at ICECAPS to associated dynamics over the GIS. Understanding how the precipitation is advected over the GIS is important in illuminating what processes affect the mass balance. Since ICECAPS is a point source, we can use observations in

concert with reanalysis data to illustrate what is occurring over the GIS regionally. We examine surface and upper level patterns, as well as back-trajectory calculations of the air masses through use of reanalysis products.

The ERA-Interim is a global reanalysis product provided by the European Centre for Medium-Range Weather Forecasts (ECMWF; Dee et al., 2011). The ERA-Interim spans the past 38 years and

has surface and pressure level profile data four times daily (0, 6, 12, and 18 UTC) with spatial resolution of 0.75° latitude and longitude. In Sect. 5, we use mean surface winds and sea level pressures for specific cases as well as calculate anomalies based on the 38-year history. We also use the ERA-Interim to examine upper-level mean winds and geopotential heights and their respective anomalies.

Back-trajectories were calculated for air masses during snowfall events at Summit. Calculations were obtained using the Air Research Laboratory's Hybrid Single-Particle Lagrangian Integrated Trajectory (HYSPLIT) model, which computes simple air parcel back-trajectories to determine the origin of an air mass for a specified time range and location (Stein et al., 2015). HYSPLIT enables the visualization of the air as it moves towards Summit Station as well as the vertical motions. We created

HYSLPLIT back-trajectories with gridded meteorological output from the National Centers for Environmental Prediction (NCEP)/National Center for Atmospheric Research (NCAR) Reanalysis Project (Kalnay et al., 1996). The NCEP/NCAR reanalysis project incorporates data from 1948 through present, with a frequency output of every 6 hours, with global coverage at a spatial resolution of 2.5°, and with 17 vertical pressure levels.



## 3 MWR-based snow classification tool

Microwave radiances have differing sensitivity as a function of frequency to different atmospheric components. For ground-based MWRs, the observed signals at all frequencies include contributions from gases like water vapour and oxygen as well as from clouds (when clouds exist in the field of view
of the radiometer). The emission from the gases is in the form of absorption lines, such as individual water vapour lines at 22.2 GHz and 183.3 GHz, or as a cluster of many absorption lines, such as for oxygen between 51.0 and 60.0 GHz. The spectral regions between these gaseous absorption features are referred to as "windows", where the contribution from the gases is relatively small. Thus, radiometer channels in these spectral windows will have a larger radiance contribution from clouds than
channels situated on gaseous absorption features. For example, in Fig. 1 (Panel a), simulations of the optical depth (OD) of the atmospheric components are shown as a function of microwave frequency. The 23.84 GHz channel is in a water vapour absorption line and thus measures a higher OD from the water vapour contribution than the neighbouring 31.40 GHz channel. The 31.40 GHz channel is not in an absorption band for either the water vapour (cyan line) or the dry gases (grey line) and is therefore
considered a window channel. The 150 GHz MWR channel is also considered a window channel, as it is free of absorption/emission bands from gases, similar to the 31.40 GHz channel (Fig. 1, Panel a). Throughout this work, we designate the 150 GHz window channel as "HF" and the 31.40 GHz window channel as "LF".

      In contrast to gas absorption, condensed cloud liquid water (CLW) exhibits continuum
absorption with much smaller spectral variation. When CLW is present in the column all channels observe emission from the condensed water, increasing the observed BT. Figure 1, Panel a, illustrates that the OD of the CLW grows larger as a function of higher MWR frequency and therefore the 150 GHz channel is more sensitive and measures about 10 times the OD from CLW as compared to the 31.40 GHz channel. When ice hydrometeors are present in the atmosphere, they will affect the observed
downwelling radiance at the surface in two ways: emission of radiation from the ice hydrometeors themselves and scattering of the surface radiation back to the MWR. In the HF (150 GHz) MWR channel, the ice hydrometeors have a high single scatter albedo of about 0.9 (e.g. Liu, 2008), which suggests that scattered radiation the dominant effect. The extinction OD from frozen water, in the form





of ice hydrometers, also has a broad continuum shape. We introduce a novel use of the ground-based MWRs to isolate IC snowfall from CLW containing snowfall by employing the ratios of the spectral response from the HF and LF window channels.

### 3.1 Spectral Response from LF and HF "window" channels during snowfall

Kneifel et al., 2010 and Pettersen et al., 2016, observed that ice falling in the column scatters the upwelling radiation from the ground back to the MWRs and results in enhanced BTs in the HF MWR channels. Thus, while the LF (31.40 GHz) MWR is insensitive to the ice hydrometeors in the column (Johnson et al., 2012), the HF MWR channels observe an enhanced BT signature from ice. The enhanced BT is due to the differences in the size parameter, which is the ratio of the hydrometeor size

with respect to wavelength. We use ratios of the observed BTs from the HF and the LF window channel to classify the snowfall by events that are coincident with clouds containing CLW and those that are ice only. Kneifel et al. (2011) and Pettersen et al. (2016) used the MWR retrieved PWV and LWP values in a radiative transfer model to simulate the BT contributions of the gas and CLW. These contributions were subtracted from the measured BT to isolate the enhanced ice signal in the HF MWR

channels. Pettersen et al. (2016) found that the MWR LWP retrievals often did not converge during snowfall events at Summit, or were biased high due to ice-enhanced BT in the HF MWR channels. Therefore, we do not use any retrievals or modelling of the CLW in this work. Figure 1, Panel b, illustrates this ratio approach with three scenarios and the accompanying response from the MWR LF and HF channels.

In clear sky situations (Fig. 1, Panel b), both the LF and HF MWR channels measure small and quickly varying BTs. The fast variations are due to measurement noise, which is uncorrelated in the two channels. Both radiometers are primarily measuring the cosmic microwave background radiation from space with small contributions from dry gases and water vapour in the column. In the second example, there is a mixed-phase cloud with supercooled CLW overhead, and both the LF and HF MWR channels

measure a higher BT signature and show similar patterns of amplitude as a function of time. This signature is due to the emission of the CLW as a function of frequency, depicted in Fig. 1, Panel a. In the final scenario, we present observations from a fully-glaciated ice cloud and there is a markedly



different response in the HF channel as compared to the LF: The LF MWR channel shows a similar pattern to that of clear sky as it is insensitive to the ice in the column. The HF channel, however, observes a large BT signature during the time that the ice cloud and precipitation is occurring. By using the differences in the ratios of the HF to LF MWR observations of each scenario, we can, with a high

degree of confidence, classify the snow into categories: precipitation originating from a fully glaciated ice cloud, i.e., "ice cloud (IC)" snow, precipitation originating from a mixed-phase cloud – snow that is has some CLW layers present, i.e., "CLW containing" snow, and precipitation that we cannot distinguish accurately the cloud type, i.e., "Indeterminate snow."

**3.2 Application of MWR classification tool to the ICECAPS dataset**

We apply the classification method to the entire 5-year dataset for the ICECAPS MWRs. We first identify the times of precipitation using the POSS power units, as the POSS is the best indicator that ice hydrometeors reached the surface without evaporating (the POSS is located within a few meters of the surface and within 10 m of the MWRs; Shupe et al., 2013). However, the POSS data is susceptible to contamination from blowing snow events. We evaluated cases of blowing snow, confirmed by observer

reports, wind speeds, and the MMCR spectral width, and determined that a threshold of 2 POSS power units is appropriate to identify precipitation events while excluding false positives from blowing snow. For all times when precipitation was identified, we use the available observations for the 31.40 and 150 GHz MWR channels from July 2010 through the end of 2015, and convert to ΔBTs as described in Sect. 2.2.1. The ΔBTs are composited for all of the precipitation events and the results are shown in Fig. 2.

20         The ratios of the composited ΔBTs in the HF and LF channels determine if the snow event is a product of a fully glaciated ice cloud, i.e., IC snow, or if there is one or more layers of supercooled CLW in the column, i.e., CLW snow. Figure 2 is annotated to illustrate the regions of the different snow types as determined by the MWR classification method. The IC snow cases are the group of points in the left lobe, where there is a strong response in the HF and minimal signal in the LF channel.

These IC snowfall events are depicted with the black arrow and are to the left of the purple, dashed line. This line is empirically determined by the HF to LF ratio response of the ice versus the CLW in the column and is used to separate the two regimes. For the cases where the snowfall is coincident with





CLW in the atmosphere, the HF and LF MWR channels both measure a BT response and the slope is lower, resulting in the right lobe of points in Fig. 2. The CLW snowfall events are denoted with a blue arrow.

There are snowfall events, which are of indeterminate type, as shown in Fig. 2 in the outlined

cyan box. The indeterminate region was calculated using multiple clear-sky days from a range of seasons and temperatures to look at the variance from computing the ΔBTs. The variation of this method may arise from environmental changes that occur between the 12-hourly radiosonde profiles. By using events categorized as clear sky from MMCR observations, we composited the HF and LF ΔBTs by season. Under clear-sky conditions, the ΔBTs maximum range for the MWR window

channels was 0.5 K (0.5 K) and 2.5 K (4 K) for the LF and for the HF for September through May months (during June, July, and August; JJA). Snowfall events that have associated BTs that are less than 2.5 K (4 K for JJA) in the HF and 0.5 K in the LF MWR channels cannot be unambiguously assigned to either IC or CLW snow and these events are therefore classified as indeterminate. This occurs when the conditions do not produce a total column amount of ice or liquid that is large enough to

produce a measureable signal over the clear-sky modelled "background".

We can now apply the MWR snow classification tool to concurrent observations from various instruments in the ICECAPS suite as well as available surface meteorological data and reanalysis products. This allows for better understanding of the different snow types through: characterizing the general cloud and precipitation properties, obtaining thermodynamic surface and profile information,

and illustrating the large-scale surface and upper-level dynamic processes. Section 4 examines the coincident measurements and retrievals available at Summit Station, while Sect. 5 explores the large-scale dynamics and implications for regional impacts over the central GIS.

**4 Characterization of snow types as observed by ICECAPS**

Figure 3 is similar to the data illustrated in Fig. 2 as it shows a two-dimensional histogram of the HF

and LF MWR ΔBTs for all precipitation events from July 2010 through 2015, however divided into summer (May through September; Panel a) and winter (October through April; (Panel b). Again, it is worth noting that the precipitation partitions into two lobes – the steep HF to LF ratio indicating the IC



process snow, and the lower slope mixed-phase process CLW associated snow. The summers have many events in both snow types, while the IC snow dominates the winters. This section will use concurrent observations and retrieved properties from the POSS, MMCR, and IcePIC instruments to quantify and characterize events within each of the snow classifications.

## 5   4.1 Occurrence and Accumulation Statistics

Figure 4 depicts the POSS-determined occurrence (Panel a) and accumulation (Panel b) statistics throughout the year. Occurrence was estimated using the POSS power threshold detection of precipitation outlined in Sect 3.2 and the associated accumulation was calculated using the Shephard and Joe (2008) algorithm for snow LWE in millimetres. All of the data are shown in percentages for all

available coincident POSS and MWR observations from July 2010 through the end of 2015 (and accounting for any instrument down time in a given month). Overall, the trend of precipitation occurrence and accumulation are similar, with slightly higher IC accumulation per event and lower indeterminate accumulation per event. By occurrence, the IC snow is 31.5 %, CLW is 48.5 %, and indeterminate is 20 % of the time, and by accumulation the IC snow contributes 35 %, CLW associated

snow is about 51 %, and the indeterminate snow is 14 %. The indeterminate snow is a small fraction of the accumulation at Summit and we will therefore focus the remaining work on the IC and CLW snowfall events.

Similar to previous studies of precipitation at Summit (Castellani et al., 2015), we find that both the occurrence and accumulation of snow is higher in mid-summer through early autumn. The largest

accumulated snowfall period is during July, August, and September comprising over 50 % of the cumulative snowfall annually, with each month contributing 15 % or more to the annual total. The peak month for snowfall accumulation is August, with ~22 %. CLW snowfall tends to increase starting in May and peaks in July for occurrence and accumulation, and falls off rapidly after September. The IC snowfall increases throughout the summer, peaks in September, and continues to have significant mass

contributions in the late fall with ~8 % of total annual accumulation during October and November. Small accumulations of IC snowfall are seen throughout the winter and spring in larger amounts than the CLW snow, and account for the majority of the accumulation deposited at Summit Station outside





the summer season. Figure 4, Panel c, shows the POSS LWE snow rate (mm hr$^{-1}$) as a function of snow classification by month in a box and whisker plot: means (horizontal line), 25$^{th}$ to 75$^{th}$ percentiles (box), and 5$^{th}$ to 95$^{th}$ percentiles (vertical line).  In general, the 25$^{th}$ to 75$^{th}$ percentile precipitation rates for the IC and CLW snow overlap, however, for every month except June and May, the IC snow has a higher

average and maximum values of POSS snow rates.  The indeterminate snow cases are largely associated with weaker precipitation rates, especially in the higher snowfall months of June through November. Overall, the majority of the accumulation deposited at Summit is from light precipitation events, with 75 % of the precipitation occurring from rates less than 0.2 mm hr$^{-1}$.

**4.2 Relationship of PWV to snowfall types**

Figure 5 illustrates the MWR retrieved values of PWV as a function of month in box and whisker plots for all available data.  Periods with snowfall at Summit have higher coincident values of PWV as compared to the distribution for all times at Summit (see Fig. 5, Panel a).  This indicates that the PWV is generally larger than the background state when there is precipitation at Summit, regardless of snow category.  The PWV values peak in July/August for both all times and precipitating times and follow a

general trend correlated to the surface temperatures.

In Panel b of Fig. 5, the monthly annual averages of the PWV are shown for each snow category as designated by the MWR snow classification tool.  For the majority of the months, the IC and CLW containing snow have similar PWV values, while the indeterminate snow has a slightly lower associated PWV. However, for most months the 95$^{th}$ percentile of the PWVs for the CLW snowfall is larger,

indicating that there are more extreme PWV values coincident with these events.  Figure 5, Panel c, shows the snow rate determined by the POSS (mm hr$^{-1}$) scaled by the corresponding retrieved PWV (in mm), which yields an approximation of the conversion rate of PWV into precipitation.  Again, the CLW and IC snow have similar values for a given month, which suggests that the CLW associated snow processes are not more or less efficient than the IC snow processes.  Thus, the differences in

accumulation observed for a given snowfall type, are largely due to differences in the fractional occurrence frequency of the regime, not because of significant differences in the PWV.  However, for all snowfall types, October through April is more efficient at turning available PWV into precipitation.





From May through September, there is much more PWV in general – coinciding with the warmer temperatures – but less snow is deposited when scaled to the PWV. This annual pattern indicates that when PWV is available during the colder and drier months, it is capable of producing relatively more snowfall when compared to the warmer summer months.

## 4.3 Radar and ice particle observations

We use the MMCR reflectivity and mean Doppler velocity observations to derive features of the vertical structure of the cloud and precipitation for the IC and CLW snow categories. We also look at retrieved properties from the MMCR of LWE snow rate, $Z_{PATH}$ (analogous to ice water path), and cloud geometric thickness (Z depth) and superimpose these on their associated ratios of the HF and LF MWR channel observations. Finally, we add some qualitative information from IcePIC photographs gathered by scientific personnel during distinct IC and CLW snow events. All of this remotely-sensed and in-situ information aids in building a more complete picture of each of the snow types and their defining characteristics.

Figure 6 illustrates vertical profile characteristics of the IC and CLW snow through MMCR reflectivities and mean Doppler velocities. All of the identified IC and CLW events are composited and corresponding MMCR properties are shown as two-dimensional histograms of the measurement as a function of height. The profiles of reflectivity for the IC precipitation cases are very deep, often 5 km or more, and have a narrow range of reflectivities for a given height, with peak reflectivity of ~15 dBZ. Panel c shows the Doppler velocities for the IC snow and again has a narrow profile, which indicates that there is ice falling and growing throughout the column as the velocities get larger closer to the ground. The reflectivity and Doppler velocity profiles for the IC snow events illustrate classic indicators of ice hydrometeor growth from the top of a cloud to the ground (Pruppacher and Klett, 2012).

Figure 6, Panel b and d show the respective two-dimensional histograms for CLW snow events. The CLW snow is associated with shallower clouds, often below 3 km, and a broader range of reflectivities, especially in the upper region of the clouds (between 1.5 to 3 km), with a similar reflectivity maximum of ~15 dBZ. The broader distribution of reflectivity may be due to the pulsed





nature of the mixed-phase clouds, as ice growth co-varies with in-cloud dynamics driven by the radiative cooling from the CLW droplets at the top of the cloud. Additionally, the CLW cases coincide with broader and weaker Doppler velocities in the lowest 2 km as compared to the IC cases. This feature could be caused by CLW indirectly as efficient cloud top cooling from the CLW droplets drives

turbulent vertical motions throughout the cloud. The weaker mean Doppler velocities may also be due to the ice habit associated with the CLW snow, i.e., particles with larger surface area such as dendrites have slower fall speeds. These characteristics observed by the MMCR for the CLW cases are consistent with features seen with shallow mixed-phase stratocumulus (Shupe et al., 2008; Verlinde et al., 2007).

We calculated three retrieved parameters from the MMCR to better understand the physical

properties of the IC and CLW snow events. We use the MMCR $Z_e$ to snow rate calculations outlined in Sect. 2.1.2 to get a LWE mass value (these values differ from the POSS snow rate, as we use a different $Z_e$ to snow rate relationship appropriate for the wavelength of the MMCR; Matrosov, 2007). The $Z_{PATH}$ is a useful alternative for ice water path but does not use conversions that are sensitive to particle size distribution and ice habit (Pettersen et al., 2016; Kulie et al., 2010). Finally, we calculate the depth of

the cloud profile as a geometric thickness ($\Delta Z$), from the MMCR. All of the retrievals are used to differentiate characteristics of the IC from the CLW snow.

Figure 7 shows the HF and LF MWR $\Delta BTs$ as two-dimensional histograms as a function of season (summer and winter), similar to Fig. 3. However, instead of binning the histogram by counts, the colour scales are the mean values of the MMCR properties associated with the $\Delta BT$ ratios. Panels a,

b, and c, depict the retrieved values for the summer season: LWE snow rate (mm hr$^{-1}$), $Z_{PATH}$ (mm$^6$ m$^{-2}$), and geometric cloud thickness (km), respectively; while Panels d, e, and f are the corresponding winter values. The MMCR snow rate for both the summer and winter is noticeably higher during the IC snow events, which is consistent with the monthly POSS-derived LWE snow rates (see Fig. 4, Panel c). The $Z_{PATH}$, which is log-binned, is consistently an order of magnitude higher during the IC snowfall

versus the CLW in both the summer and winter. The clouds tend to be geometrically thicker during the IC events while the CLW cases are geometrically thinner.

In general, the retrieved properties obtained from the MMCR yield consistent conclusions as the MMCR reflectivity and Doppler velocity observations: The IC snow events are associated with deep





systems with ice falling from the very top of the cloud and growing throughout the column. Although they are less common, the strongest IC snow events have higher potential mass deposition as evidenced by the correlated high snow rate and $Z_{PATH}$ values. The CLW cases tend to be shallower with evidence of supercooled CLW at the top of the cloud, have lower $Z_{PATH}$, and slightly less deposition per event,

though they occur more frequently.

We looked at IcePIC photos during identified IC and CLW snowfall cases. Local scientific personnel gathered ice hydrometeors sporadically to provide qualitative evidence of differences in ice habit. Some example IcePIC photos for specific events from each category of snowfall are highlighted in Fig. 8. For all the cases that were unambiguously correlated with an IC snow event, the ice habits

observed are mostly bullets, bullet rosettes (of many number branches), and some columns and small plates (Fig. 8, left). This provides additional evidence that the IC snow events have ice originating at the top of the cloud growing throughout the column, as these habits are indicative of very cold and pristine conditions devoid of CLW (Korolev et al., 1999). The IcePIC photos taken during CLW snow events yielded mostly dendrites and sectored plates with occasional small amount of riming, which is

consistent with ice falling through CLW layers and warmer temperatures (Fig. 8, right). It is worth noting that variability in the ice habit and the particle size distribution of the snowfall does impact radar reflectivity to snowfall relationships. Studies show that different particle size distribution and ice habit can impact the calculated snow rate from reflectivity for both the POSS and MMCR frequencies (Liu, 2008; Dolan and Rutledge, 2009; Kulie and Bennartz, 2009). Though we do have some evidence of

differing ice habits for the IC and CLW precipitation, we do not have any particle size distribution information and cannot adjust the radar to snow rate based on the snow category. Therefore we are using a generalized, average relationship for all snow categories to acquire snow rate and accumulation information from both the POSS and MMCR.

## 5 Source air mass characteristics and dynamics associated with the snow types

In this section we explore the origins of the air masses and their associated dynamics for both IC and CLW snow events. First, the dynamics can help explain why half the precipitation events are associated with mixed-phase clouds with layer(s) of supercooled CLW, while another 35 % are coupled to deep,



fully glaciated ice clouds. We find that there are distinct differences in the air mass behaviours for either type: The IC snow events propagate quickly over the southeast region of the ice sheet, have very deep layers of water vapour, and are likely advected over the GIS through large-scale vertical motion associated with the regional meteorology and topography, but may have less small-scale vertical motion

(turbulence). The CLW events advect slowly across the southwest and southern portions of the GIS, tend to be shallow, and follow a quiescent flow to Summit. The CLW cases have calmer large-scale motion of the air mass, but much more small-scale turbulence driven by the CLW itself, which is consistent with characteristics of persistent Arctic mixed-phase clouds (Shupe et al., 2008; Morrison et al., 2012). Secondly, by understanding how the precipitation gets to Summit through the large-scale

dynamics, we explain what is occurring regionally and, therefore gain broader knowledge of how the point observations at Summit Station apply to the central GIS.

### 5.1 Surface winds at Summit

Figure 9 (left) shows the total surface topography of Greenland (contoured from sea level to 3100 MASL), which includes the bedrock and recent measurements of the ice surface from the IceBridge

campaign (Morlighem et al., 2015). The location of Summit Station is nearly at the top of the GIS (indicated with purple circle) and is both far from the ocean (400 km from the east and west coastlines and over 1000 km from the southwest and southeast). Therefore, understanding from where the air masses originate helps in illuminating how the precipitation arrives at Summit. We first look at the 10-meter surface winds (National Oceanic and Atmospheric Administration, Global Monitoring Division)

coinciding with the IC and CLW snow events. Figure 9, Panel a, shows the wind speeds and directions for all dates and times from mid 2010 through 2015 for Summit. In general, precipitation occurs at Summit when the surface winds originate from south (though north winds do occur, they rarely bring precipitation) and these winds are often stronger than the mean winds (Fig. 9, Panels b and c).

By examining the coincident IC snowfall surface wind speeds and wind directions (Fig. 9, Panel

b), we see that there is a preference of these events to originate from the southeast direction, however there is a distributed mode to the south and southwest as well. The IC snow event winds are much stronger than the mean state winds for all times at Summit, with most cases having winds stronger than




9 m s$^{-1}$. This is interesting as the majority of snow accumulation in Greenland is along the southeast coastal mountain range, and the ocean to the immediate southeast is a region with one of the highest occurrence snowfall locations in the Northern Hemisphere (Hanna et al., 2006; Kulie et al., 2016). However, much of this snowfall does not make it up and over the steep orography along the southeast

coast of Greenland to the central GIS (Hanna et al., 2006). The direction and strength of the surface winds associated with the IC snowfall indicate that strong dynamics may be able to advect water vapour and precipitation-rich air masses from the southeast coastal region atop the central GIS.

When considering the mixed-phase CLW containing snowfall cases, the winds are predominately coming from the west-southwest to south-southwest directions (Fig. 9, Panel c). Recent

studies of long-lived mixed-phase clouds at Summit show that they originate equivalently from the west, south, and east (Edwards-Opperman et al., submitted), however many of these clouds are either not precipitating or are precipitating below the POSS detection threshold (outlined in Sect. 3.2), and therefore only a subset are included in this work. Though there is a broader range of surface winds coincident with the CLW snowfall cases, the majority are coming from a different direction when

compared to the IC snow, with 70 % originating from the west to the south of Summit Station (though there is a small amount originating from the southeast). These winds are not as strong as the wind speeds seen with the IC snow cases, but they are faster than the average winds seen for all times at Summit. This is consistent with previous studies, which showed that most clouds (of which the majority are mixed-phase and contain layer(s) of CLW) and precipitation occur under winds with

southern and south-westerly flow (Shupe et al., 2013; Castellani at al., 2015). The surface winds indicate that these air masses are traveling slowly up the comparatively gentle slope southwest of Summit.

## 5.2 Regional meteorological conditions for snow type

In addition to the local meteorological conditions at Summit, we examined the regional surface patterns

and large-scale dynamics associated with each snowfall regime using the ERA Interim Reanalysis. In general, it has been shown that precipitation over the central GIS is associated with moisture coming from the south via onshore and upslope flow (Bromwich et al., 1998; Hanna et al., 2006; Schuenemann





et al., 2009).    We use the mean and climatological anomalies of sea level pressures and surface (10 meter) winds, as well as the 500 mb geopotential heights and upper-level winds to infer how the precipitating air masses get to Summit and what processes may glaciate the clouds as opposed to sustain layer(s) of CLW.  Previous sections of this study included all identified IC and CLW snowfall events,

regardless of their duration.  However, since the ERA Interim Reanalysis product has a four times daily resolution (at 00, 06, 12, and 18 UTC) we wanted to include only those events long enough to say with confidence that they occurred for most of an hour and at a time near the reanalysis product.  We filtered the snow cases and used events that were duration of minimum of 45 minutes of an hour and within 2 hours of an ERA Reanalysis time step.  We did not allow for more than one value in the same day

unless 12 hours or longer apart to avoid one storm biasing the results.  This method was purposefully conservative and yielded 90 IC and 84 CLW snowfall cases.  The majority of the IC snow cases are from August through November, and all of the CLW snow cases are in May through September.  To calculate the anomalies we used the 38-year dataset of surface and pressure level values and averaged these into monthly means for each longitude and latitude used in our study.  Anomalies were then

calculated by subtracting these historical monthly means from specific cases of identified IC or CLW snowfall.

 We first look at the mean sea level pressure (SLP) patterns in the region around Greenland for the composited IC snow events.  Figure 10, Panel a, shows an extremely deep low-pressure feature (sea level pressure < 1000 hPa) wrapping around the horn of southern Greenland.  This cyclonic feature has

accompanying strong winds that originate from northern Canada and circulate counter clockwise, eventually towards the southeast coast of Greenland, with surface winds at Summit from the southeast. The SLP and wind anomalies for the IC snow cases are shown in Fig. 10, Panel c.  There is a negative SLP anomaly coincident with the location of the centre of the cyclone, and an anomalously high SLP anti-cyclonic feature to the east of Greenland.  Previous work on synoptic forcing of precipitation over

the GIS by Schuenemann et al. (2009) showed a similar pattern of coupled low/high SLP anomalies generated precipitation both over Greenland and over the central GIS.  The cyclone feature near the horn of Greenland is potentially a product of lee cyclogenesis, as it forms in the lee of the topographic ridge along the southern tip of Greenland (Rogers et al., 2004; Schuenemann et al., 2009).  Greenland



lee cyclogenesis is also found to correlate with precipitation over the GIS, though most strongly in the southern region (Bromwich et al., 1998; Schuenemann et al., 2009).

In addition to the surface products, we examine the 500 mb geopotential height and wind patterns, both the means and the anomalies (see Fig. 11). For the IC snow events, the mean

geopotential heights show a strong trough and ridge feature centred along the long axis of Greenland. The upper-level mean winds follow this height structure and show advection from the southeast coast to Summit Station. Just to the east of the trough is an area of upper-level divergence that creates strong vertical ascent throughout the column (Holton, 2004). This is located over a region of the North Atlantic Ocean with very high occurrence of snowfall and accumulation along the Greenland coast

(Bromwich et al., 1998; Hanna et al., 2006; Kulie et al., 2016). Together, these features indicate that large-scale lifting likely pulls precipitation and water vapour from low in the troposphere up into the column and the upper-level winds then push this deep precipitation over the steep topography and onto the central GIS. Figure 11, Panel c, show the 500 mb geopotential height and wind anomalies for the IC snow events. Similar to the anomalies in the SLP analysis, the IC snow events have a dipole structure

centred over Greenland, with lower than average heights to the west and a much higher than average ridge feature to the east. The upper-level wind anomalies are originating from the southeast and are strong compared to the mean state winds. The dynamics implied by the 500 mb mean geopotential heights and anomalies support the deep, characteristic ice clouds observed by the MMCR at Summit. Additionally, according to the reanalysis temperature and relative humidity profile, the column is

saturated with respect to ice up to 300 mb where the temperatures are below -40 C for the entire area over the central GIS. This indicates that ice is forming at the top of these clouds, thus adding to the evidence that they are fully glaciated systems. Once ice has formed at the top of the cloud it will start to descend, and in a water vapour rich environment it will grow and eventually precipitate out to the surface, suggesting that these systems are snowing across the southeast central GIS as they move

quickly towards Summit.

The regional mean SLPs and winds for the CLW snow cases are depicted in Fig. 10, Panel b, and show a relatively uniform pressure pattern over Greenland. The surface winds show weaker flow from the south approaching the Greenland coastline with stronger winds from the southwest at Summit



Station. The SLP anomalies for the CLW snow cases are much weaker than those seen in the IC cases (Fig. 10, Panel d). In general, there is a broad, weak anti-cyclonic anomaly over most of the GIS and to the south and southeast, with a weak cyclonic anomaly near the United Kingdom. The wind anomalies show more moisture is coming from the south and southwest when compared to the mean state, which

is consistent with studies of precipitation over the GIS (Bromwich et al., 1998; Hanna et al., 2006). The SLP anomalies shown in Fig. 10, Panel d, are consistent with calm conditions and weak forcing of vertical motions as there is a broad high SLP anomaly over most of the GIS. These features are favourable for Arctic mixed-phase clouds (Morrison et al., 2012; Shupe et al., 2006) and therefore consistent with the CLW snowfall cases.

10       The 500 mb mean geopotential heights and winds for the CLW snowfall events show a very different picture from that of the IC snow: the mean geopotential height is fairly uniform across Greenland and the upper-level winds are calm and flowing over the GIS from the south-southwest. This indicates a weak, quiescent flow that is slowly traversing up and across the GIS from the southern and south-western coasts. As stated previously, Arctic mixed-phase clouds are resilient in weakly-forced

conditions such as those illustrated in Fig. 11, Panel b, for the CLW snow events (Morrison et al., 2012). Since the CLW snow is connected to these longer lived and slower mixed-phase cloud systems, they are likely to periodically snow over the GIS on their way to Summit. Figure 11, Panel d, shows the 500 mb geopotential height and wind anomalies for the CLW snow events. The CLW snow cases show that the mostly flat 500 mb mean geopotential heights across Greenland are, on average, anomalously

high over a spatially extensive region, and even though the upper-level mean winds are fairly weak, they are anomalously strong compared to the background conditions (Fig. 11, Panel d). This is consistent with studies that have shown that higher than average 500 mb geopotential heights over Greenland are coupled to precipitation over the central GIS (Hanna et al., 2016).

### 5.3 Back-trajectories for each snow type

Using data from the NCAR/NCEP reanalysis project and the HYSPLIT modelling tool, we construct 36-hour back-trajectories for the IC and CLW events. We present results from back-trajectories of air masses at 3 km above Summit Station – although we looked at other heights, 3 km seemed to be the



best compromise to capture motions associated with each precipitation classification (top of the CLW and middle of the IC snow cases) while minimizing localized artefacts from the GIS topography. Figure 12 shows the spatial movement and vertical motions (mean and standard deviation) for IC cases (left, top and bottom) and CLW (right). The backtrajectory above ground level (AGL) values represent

the altitude above the model terrain (or ocean) height respective to the path of each trajectory. The IC snow events are mostly originating from over the North Atlantic Ocean, these air masses are moving very quickly over the GIS (with respect to the 36-hour reanalysis period), and these events are lifted a total of 5 km on average (from a mean of 1 km AGL over the ocean surface, to 3 km AGL over Summit) by the strong vertical motions off the coast of Greenland. For the CLW snowfall cases (Fig.

12, right), the back-trajectories originate to the south and southwest of Summit, these air masses are moving slower than the IC events, and the mean vertical motion is only slightly upwards, though the variance is larger with some air masses descending. In general, the HYSPLIT modelled back-trajectories confirm the dynamics that were inferred from the SLP and geopotential height maps, as well as the ICECAPS observations of cloud and precipitation properties for each snow regime.

**6 Conclusions**

We introduced a MWR-based method for classifying the precipitation at Summit to discriminate snow events originating from fully glaciated ice clouds (IC) from those associated with mixed-phase clouds (CLW). We are able to isolate IC snowfall from CLW snowfall by employing the ratios of the spectral response from the HF and LF MWR window channels. Key to this method is the HF (150 GHz) MWR

channel, which is shown to be an important tool for ground-based classification of precipitation regimes over central GIS.

Observations from ICECAPS instruments demonstrate that the CLW snow is the dominant regime of precipitation with 51 % accumulation, almost all of which occurs in the summer months. The IC snow, however, is a large component of the accumulation at Summit – accounting for about 35 % of the

total. The IC snow is the main source of accumulation during the non-summer months and is capable of producing relatively more accumulation with less available PWV. IC snow events have higher than average winds, predominately from the southeast, indicating that the events are likely coming over the





steepest part of the Greenland coast. The CLW snow events have moderate winds from the south and southwest, traversing up a gentler slope to Summit. The coincident MMCR observations for the IC snow cases show deep clouds indicative of ice growth throughout the column: the reflectivity and Doppler velocity distributions are both relatively narrow and the mean values increase as the

hydrometeors reach the surface. Contrarily, MMCR observations for the CLW snow cases illustrate shallower clouds with broader ranges of reflectivities and more frequent occurrence of lower Doppler velocities, indicating layers of supercooled CLW droplets, the shallow dynamics associated with these clouds, and different ice particle distributions.

The large-scale dynamics, as indicated by the ERA Interim reanalysis, find distinct synoptic regimes
associated with IC and CLW snow that are consistent with the observations from the instruments at Summit. The mean SLP map for the IC snow cases shows a strong low to the east of the southern tip of Greenland implying that these topographical lee cyclones are a key mechanism for air mass advection during these precipitation events (Rogers et al., 2004; Schuenemann et al., 2009). Additionally, the SLP anomaly map for the IC snow shows two low pressure anomalies – one in Baffin Bay and one wrapping
around the horn of Greenland – implying that these storms are potentially bifurcated by the Greenland ridge topography, a storm pattern which is correlated with precipitation atop the GIS (Schuenemann et al., 2009). The mean SLP map for the CLW snow cases show a calm, flat high pressure across most of Greenland. The SLP anomalies are slightly positive over much of Greenland and there is a large anticyclone feature from the southwest to the northeast over the North Atlantic correlated with the CLW
snow. The mean high SLP over Greenland promotes calm advection of mixed-phase clouds from the southwest and south up and over the central GIS, which is consistent with previous observations (Appenzeller et al., 1998; Bromwich et al., 1999).

The 500 mb geopotential mean heights and anomalies and HYSPLIT back-trajectories illustrate how precipitation is formed and how it may be affecting the central GIS. The mean 500 mb geopotential
height maps show how the IC and CLW snow regimes are advected to Summit and are consistent with the observations from the ICECAPS instruments. The mean 500 mb geopotential heights for the IC snow have a large, coupled trough and ridge feature centred over Greenland. To the east of the trough is an area of upper-level divergence, which induces large vertical updrafts throughout the column. The





IC snow is characterized by deep cloud systems where the ice can grow and precipitate out over the GIS. The evidence of upper-level divergence implies that large-scale upward motion creates low-pressure systems, which transport water vapour upwards along the south-eastern slope of Greenland. These deep systems are then advected over the central GIS and likely also precipitate over the southeast

central GIS as they travel towards Summit. The mean 500 mb geopotential heights associated with the CLW snow cases are very flat and show a region of quiescent upper-level flow. The CLW snow is associated with shallower systems with evidence of supercooled CLW at the top of the clouds. The quiescent flow, slowly advecting up and over the south and southwest Greenland topography is an environment favourable for long-lived mixed-phase clouds (Morrison et al., 2012, Shupe et al., 2006).

The vertical motions and relative speed of the air masses for each snow regime from the HYSPLIT back-trajectory analyses illustrate similar mechanisms.

These dynamics have implications for both how precipitation is formed and how it arrives at Summit. The patterns of the SLP and 500 mb geopotential height anomalies for the IC and CLW are very different. Features seen in the anomaly maps may relate to climate indices – particularly the North

Atlantic Oscillation and the Greenland Blocking Index, as both have been linked to precipitation over the central GIS (Bromwich et al., 1999; Hanna et al., 2016). The conclusions from this study warrant further work investigating the dynamics of the IC and CLW snow cases by season and comparing the resulting SLP and 500 mb geopotential height anomalies to the seasonal North Atlantic Oscillation and Greenland Blocking Index.

This study illustrates that there are two distinctive regimes of snowfall at Summit Station: snow from ice clouds and snow from mixed-phase clouds. The two identified snow classifications have dissimilar dynamics governing how the precipitation reaches the central GIS and may therefore have very different responses to a changing climate. Historically, it is found that changes in atmospheric circulations and storm systems are the dominant force for changes in precipitation over the GIS and not

increases in temperature (Kapsner et al., 1995). The distinct large-scale dynamical drivers for each snowfall type suggest potential differences in response to climate change. If these precipitation regimes respond in different ways to rapid climate change in the Arctic, the magnitude of the mass balance of the central GIS over time is highly uncertain.



## 7 Data Availability

All data used in this work is collected by the ICECAPS project and is publically available:

- MWR data at arcticdata.io: urn:uuid:d2029d7c-3843-4fda-97f2-72f194455ae8
    - dataset: doi:10.18739/A2R79T
    - dataset: doi:10.18739/A27795
    - dataset: doi:10.18739/A2ZR3K
    - dataset: doi:10.18739/A22J6K
    - dataset: doi:10.18739/A2HJ57
    - dataset: doi:10.18739/A24B8W
- MMCR data at arcticdata.io: urn:uuid:e557c419-8b55-4155-8954-2f60bb4b8c0d
    - dataset: doi:10.18739/A2G74F
    - dataset: doi:10.18739/A2M17R
    - dataset: doi:10.18739/A2BJ3X
    - dataset: doi:10.18739/A2318G
    - dataset: doi:10.18739/A2121J
- POSS data at arcticdata.io: urn:uuid:6ab14e9d-e3d5-46d2-aa3e-297afec1814d
    - dataset: doi:10.18739/A2318G
    - dataset: doi:10.18739/A2H20H
    - dataset: doi:10.18739/A2CB7P
    - dataset: doi:10.18739/A27J64
    - dataset: doi:10.18739/A27J64
    - dataset: doi:10.18739/A2VB8D
- Radiosondes data at arcticdata.io: urn:uuid:21d4e7fa-041a-4baf-aedd-38dc6d388661
    - dataset: doi:10.18739/A2F490
    - dataset: doi:10.18739/A25N4S
    - dataset: doi:10.18739/A2X508
    - dataset: doi:10.18739/A2NN44
    - dataset: doi:10.18739/A2WZ18



- o 2015 dataset: doi:10.18739/A2GZ1J
- IcePIC
  - o Available at anonymous@ftp1.esrl.noaa.gov:/psd3/arctic/summit/icepic/

A merged dataset of the snow classification product is in development. It will be available with a DOI

5    before final publication.



## 8 Appendix

This is a list of definitions for the acronyms that are most frequently referenced in this manuscript:

| Acronym | Name |
| --- | --- |
| AGL | Above Ground Level |
| BT | Brightness Temperature |
| CLW | Cloud Liquid Water |
| GIS | Greenland Ice Sheet |
| HF | High Frequency, 150 GHz channel |
| IC | Ice Cloud |
| ICECAPS | Integrated Characterization of Energy, Clouds, Atmospheric state, and Precipitation at Summit |
| LF | Low Frequency, 31.40 GHz channel |
| LWE | Liquid Water Equivalent |
| LWP | Liquid Water Path |
| MMCR | Millimetre Cloud Radar |
| MWR | Microwave Radiometer |
| POSS | Precipitation Occurrence Sensor System |
| PWV | Precipitable Water Vapour |
| SLP | Sea Level Pressure |



*Acknowledgements:* This study is supported by National Science Foundation grants Nos. PLR1304544, PLR1355654, PLR1303879, PLR1314156, PLR1304692, PLR1314358, PLR1414314, and PLR1420932. The POSS was provided by Environment and Climate Change Canada, while the NOAA Earth System Research Laboratory provided the MMCR. C. Pettersen would like to thank M. Kulie and

5   M. Breeden for their valuable feedback via personal communication. The authors would like to thank the Summit Station science technicians and staff as well as Polar Field services for their continued dedication to gathering data and maintaining instrumentation. The authors would like to thank the ACP Co-Editor and the anonymous reviewers for their helpful comments and feedback.



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



**Table 1. Subset of ICECAPS instruments used for this study (modified from Table 1 in Shupe et al., 2013)**

| Instrument Name | Specifications | Measurements | Derived Parameters |
|---|---|---|---|
| HATPRO | Frequencies: 22-32 GHz (7 channels) 51-58 GHz (7 channels) 2 to 4-second resolution | Downwelling Brightness Temperature | Precipitable water vapor |
| MWRHF | Frequencies: 90 and 150 GHz. 2 to 4-second resolution | Downwelling Brightness Temperature | Precipitable water vapor |
| MMCR | 35 GHz (Ka band), 8-mm wavelength. 45-meter vertical bin size. 2-second resolution | Reflectivity, Doppler velocity, Doppler spectral width | Cloud micro and macro-physics, cloud dynamics, precipitation rate, ice water path |
| POSS | 10.5 GHz (X band), single bin, near surface, 1-minute resolution | Reflectivity, Doppler spectra | Precipitation occurrence, and rate |
| RS-92K or RS-92SGP Radiosondes | Twice daily (00 and 12Z), 1-second resolution. | Temperature, relative humidity, pressure, winds | Cloud temperature, tropospheric thermodynamic structure |
| IcePIC | Canon D50 DSLR, 1.5 μm resolution, 6.1 megapixels | Digital photographs | Ice crystal habit, Qualitative assessment of riming, aggregation |





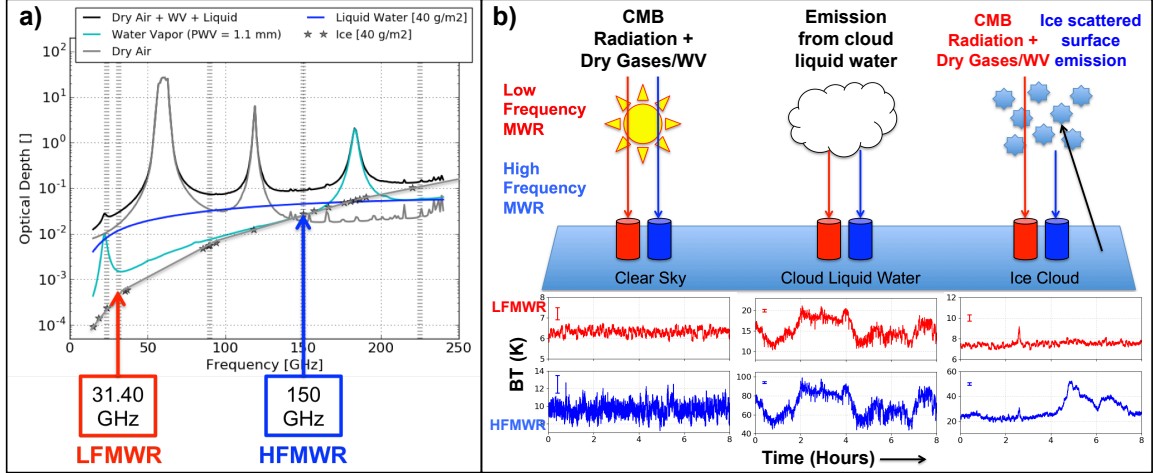

**Figure 1. Panel a (left) shows a representation of modelled extinction optical depth as a function of frequency for the atmospheric components under conditions relevant for Summit: both the liquid water path and ice water path are 40 g m$^{-2}$, and the water vapour and dry gas concentrations are from the Standard Subarctic Winter profile starting at 3 km. The red and blue arrows highlight the microwave channel observations used in the study (low and high frequencies, respectively). Note the different spectral slopes of the ice versus the liquid versus the water vapour contribution. Panel b (right) is a schematic representation of the spectral response of the low (red) and high (blue) frequency microwave radiometers under conditions of clear sky (left), cloud liquid water in the column (middle), and precipitating ice cloud (right). Error bars denoting the MWR channel measurement precision is shown in the top left corner of each plot (0.3 K and 1.0 K for the low and high frequency channels, respectively).**

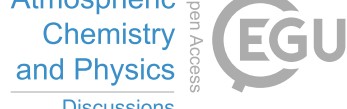

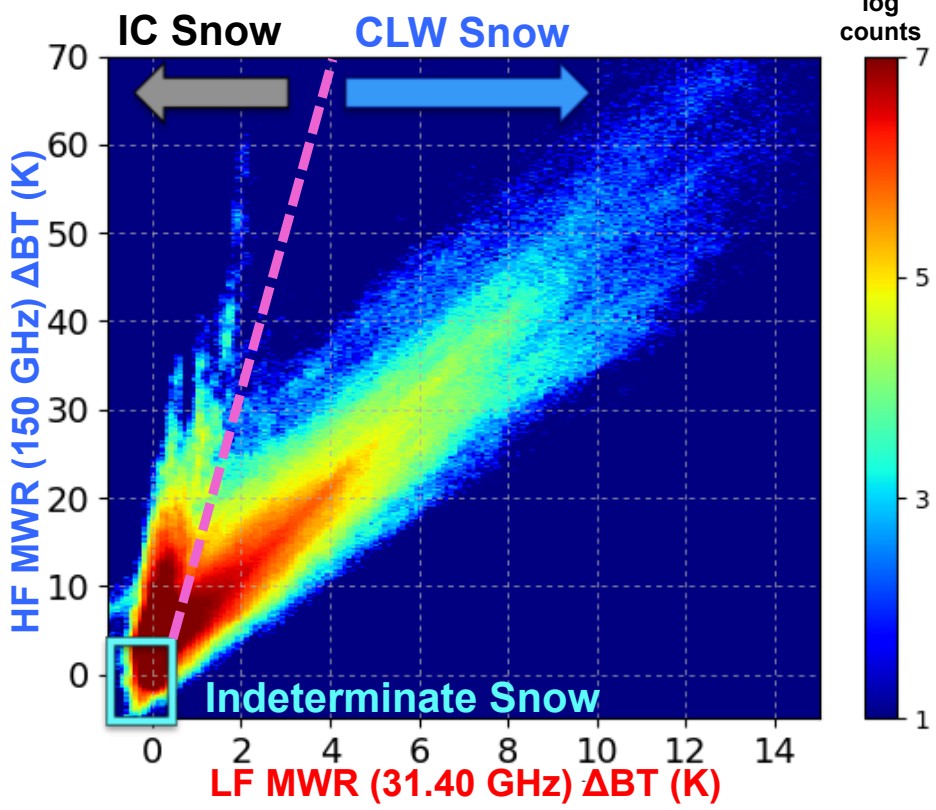

**Figure 2. All available MWR data for 2010 – 2015 during precipitation (as determined by the POSS power unit threshold). These values are delta BT where the clear sky forward model RT run is subtracted from the MWR observations. The arrow annotations show the regions of IC snow (to the left of the dashed line), snow with associated CLW in the column (to the right of the dashed line), and snow of indeterminate type (in the cyan shaded region). The indeterminate region is defined by the sample distribution in clear sky, and captures the residual variance due to uncertainties in the modelling of the gas absorption optical depth.**





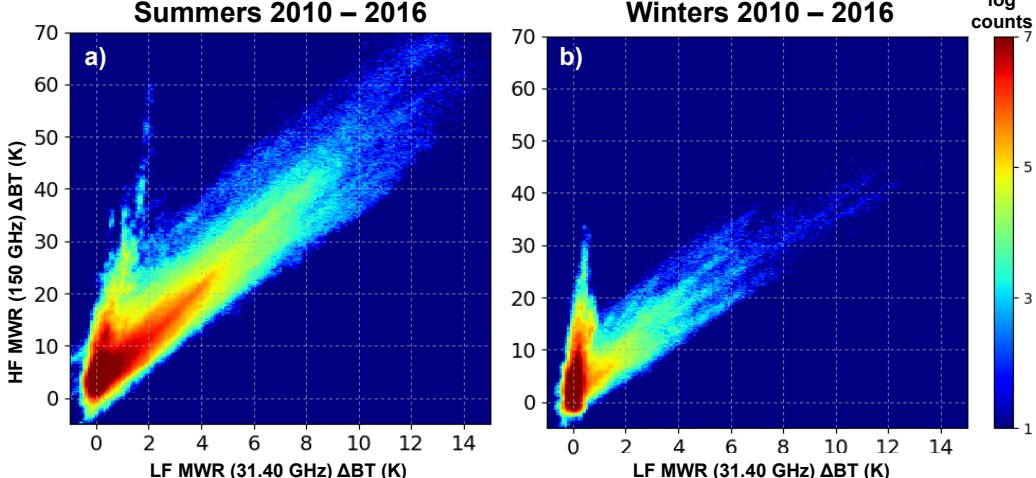

**Figure 3.** These are the MWR observations minus the clear-sky contribution, for all available data during precipitation events from 2010 to 2015. The summer precipitation is show in Panel a (left; summer is defined as May through September), and the winter precipitation is shown in Panel b (right; winter is defined as October through April). The summer MWR observations indicate both IC and CLW snow events occur through the season, though more CLW events. The winter season tends to strongly favour the IC snow events.





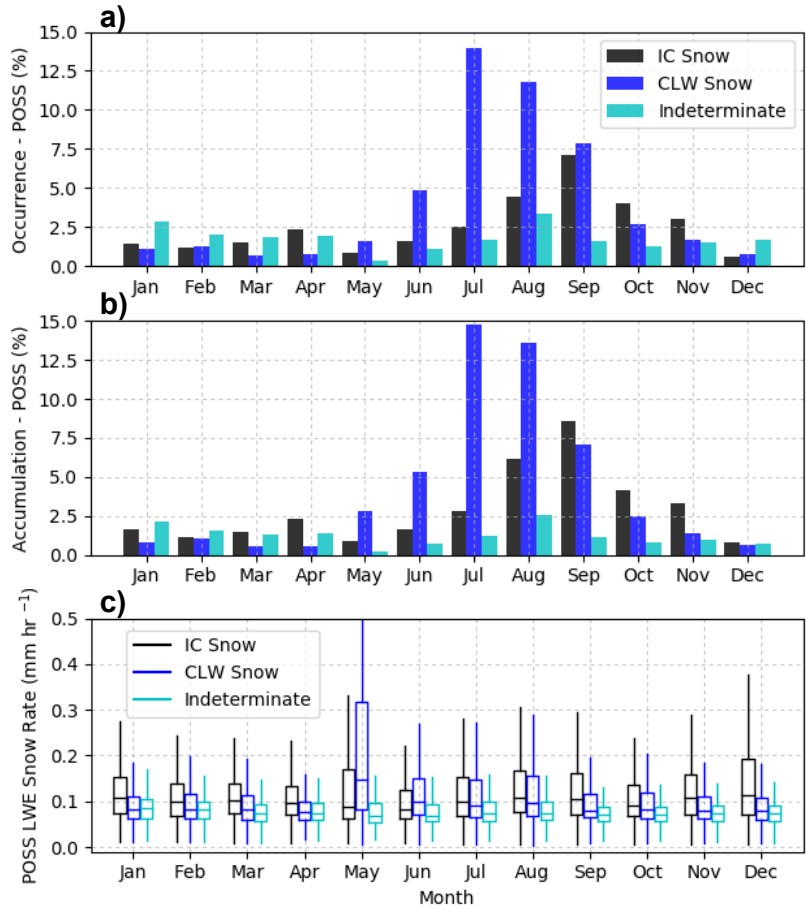

**Figure 4.  The POSS statistics from 2010 – 2015 for the MWR filtered precipitation events.  Panel a (top) shows snow amounts by Occurrence (POSS) for all data: IC – 30.5 %, CLW – 48.5 %, and Indeterminate – 21 %.  Panel b (bottom) shows snow amounts by Accumulation (POSS) for all data: IC – 35 %, CLW – 51 %, and Indeterminate – 14 %. The POSS snowfall amounts and snow rates were calculated using the Joe and Sheppard (2008) Z to S relationship.  (Note: Panel c shows very high values for the CLW snow in May, which is due to an unusually large storm dominating the results).**



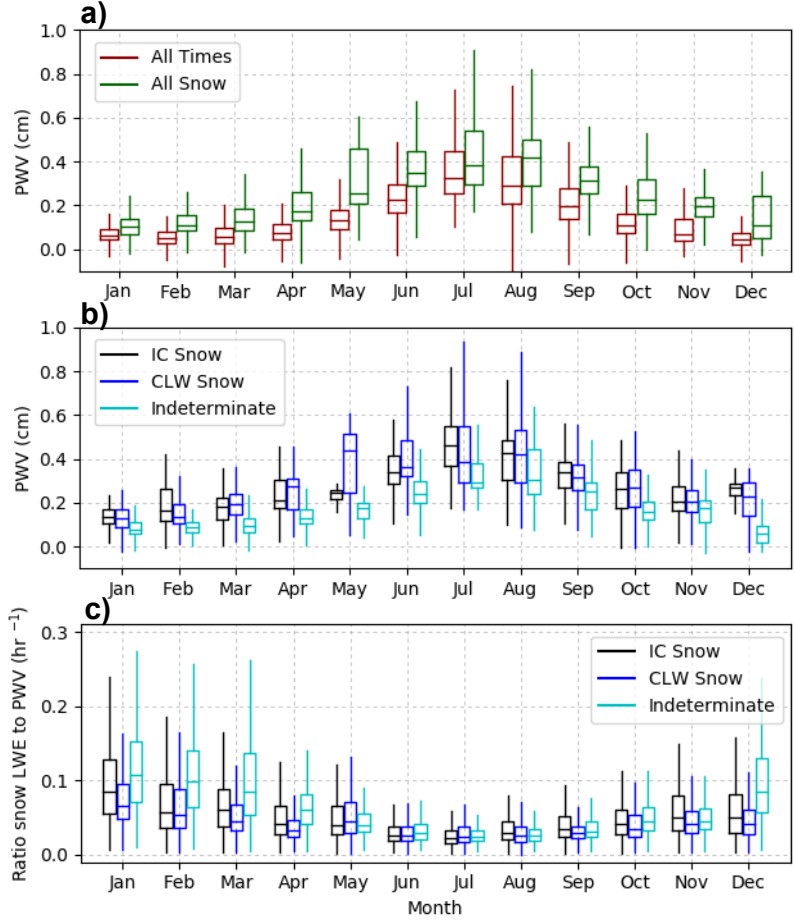

**Figure 5. Panel a shows the average annual PWV as a function of the month for MWR data from 2010 –2015. The PWV values during snowfall events, regardless of type, are higher than that of the PWV averages during all times (precipitation and non-precipitating). Panel b shows the average PWV associated with each MWR-determined type of snowfall. Panel c shows the ratio of the average snow rate measured by the POSS in LWE mm hr⁻¹ to the associated PWV in mm, thus giving a rate of how efficiently the PWV converts to precipitation for each month and snow type.**



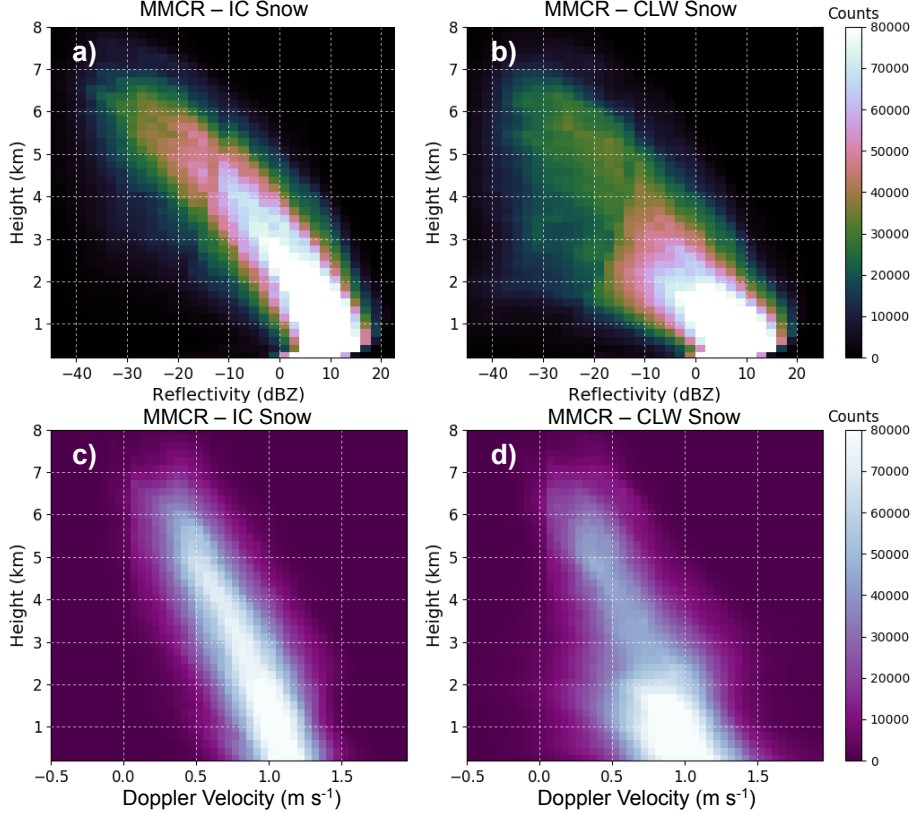

**Figure 6. Composite two-dimensional histograms of MMCR properties for each MWR-determined snow type are shown. Each histogram uses a linear colour scale with a maximum value of 80,000 counts. Panels a (top, left) and b (top, right) show the MMCR reflectivity as a function of height for all the IC and CLW snow cases, respectively. Panels c (bottom, left) and d (bottom, right) show the MMCR Doppler velocities as a function of height. These composites of the IC and CLW precipitation highlight different characteristics between the two snow modes.**





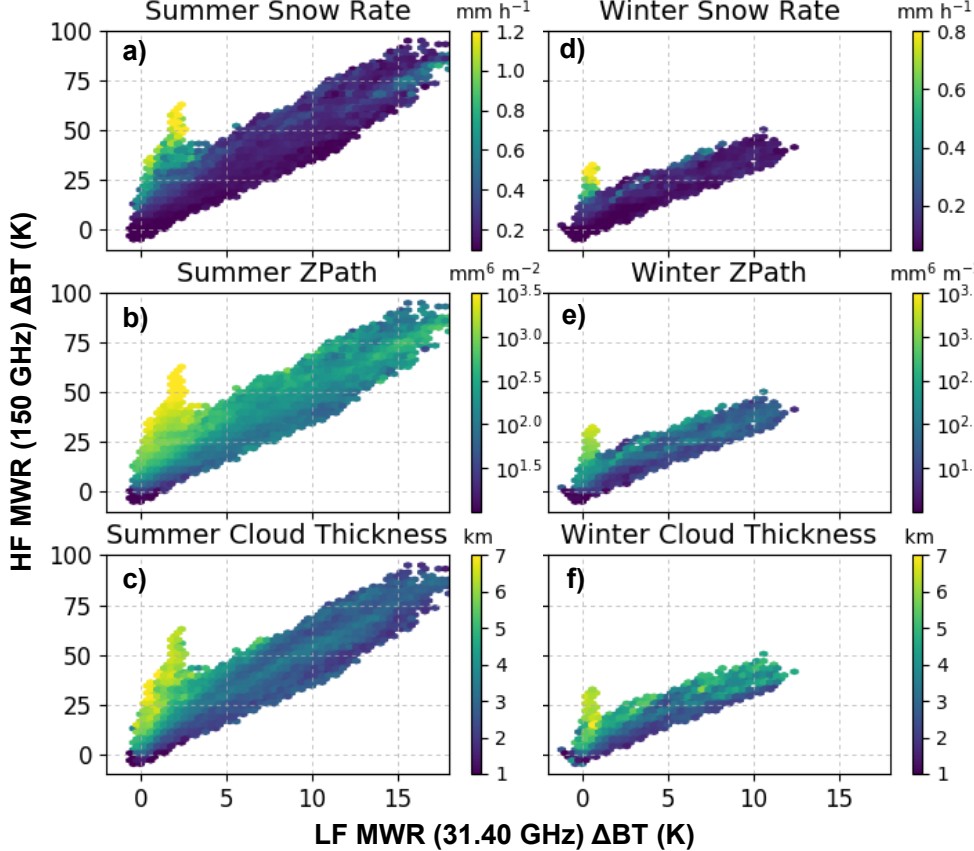

**Figure 7. Values of MMCR calculated snow rate, $Z_{PATH}$, and cloud thickness are calculated for all the precipitation events and plotted with the associated HF and LF MWR observations. The top Panels (a, b, and c) depict these characteristics for the summer months and the bottom Panels (d, e, and f) for the winter months. Regardless of season, the IC precipitation has a higher instantaneous snow rate than the CLW cases. Additionally, the $Z_{PATH}$ values for the IC snow cases are much higher than the CLW cases. And the IC snow tends to be associated with deeper clouds than the CLW snow.**





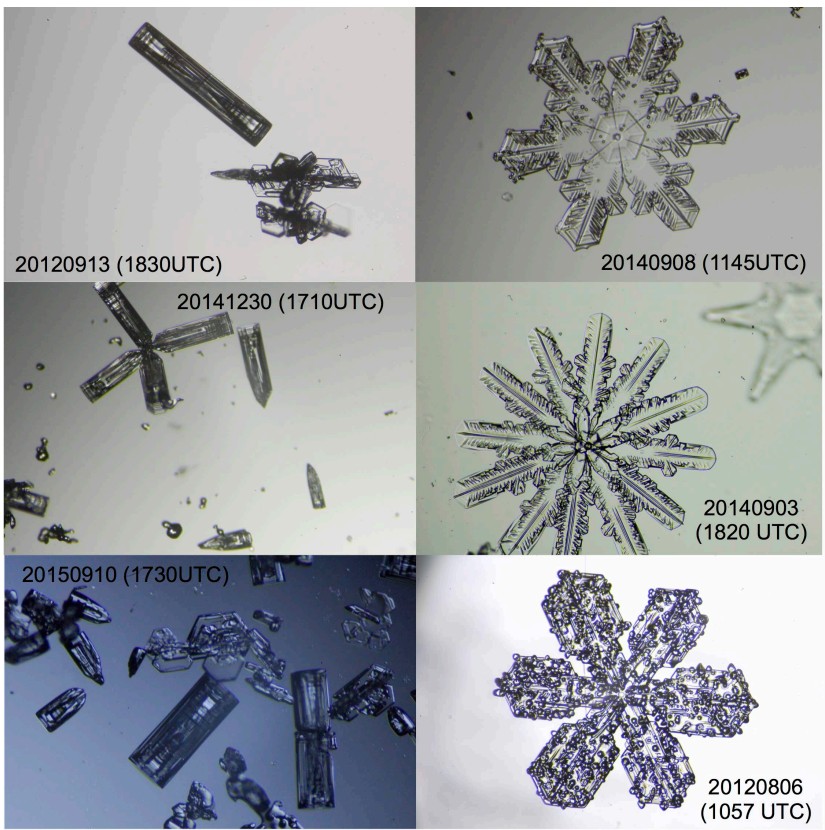

**Figure 8. Examples images from the IcePIC camera of ice cloud (IC) originating snow events (left) and for mixed-phase CLW containing snow events (right).**





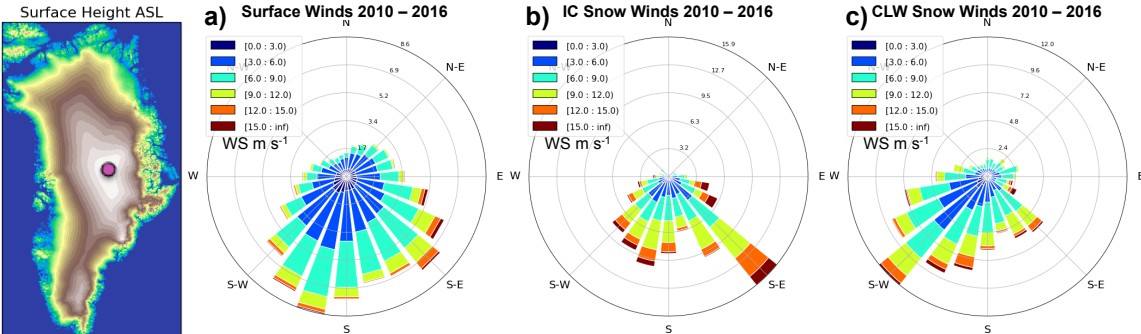

**Figure 9.** Figure showing the surface height of Greenland was created using measurements taken during the IceBridge campaign (left). The surface height combines the contributions from both the bedrock and ice sheet topography. Surface winds from the Summit NOAA meteorological data are shown. For reference, Panel a shows all surface winds from 2010 –2015 for all times. Panel b (middle) shows the surface winds for the MWR-determined IC snow cases. These winds tend to come out of the southeast with little variability and are much stronger than the average winds. Panel c (right) shows the surface winds for the MWR-determined CLW containing snow cases. Associated winds tend to be from the west to south with a maximum amount from the southwest direction. Though the CLW snow cases have stronger winds than average, they are not as strong as the winds associated with the IC snow.



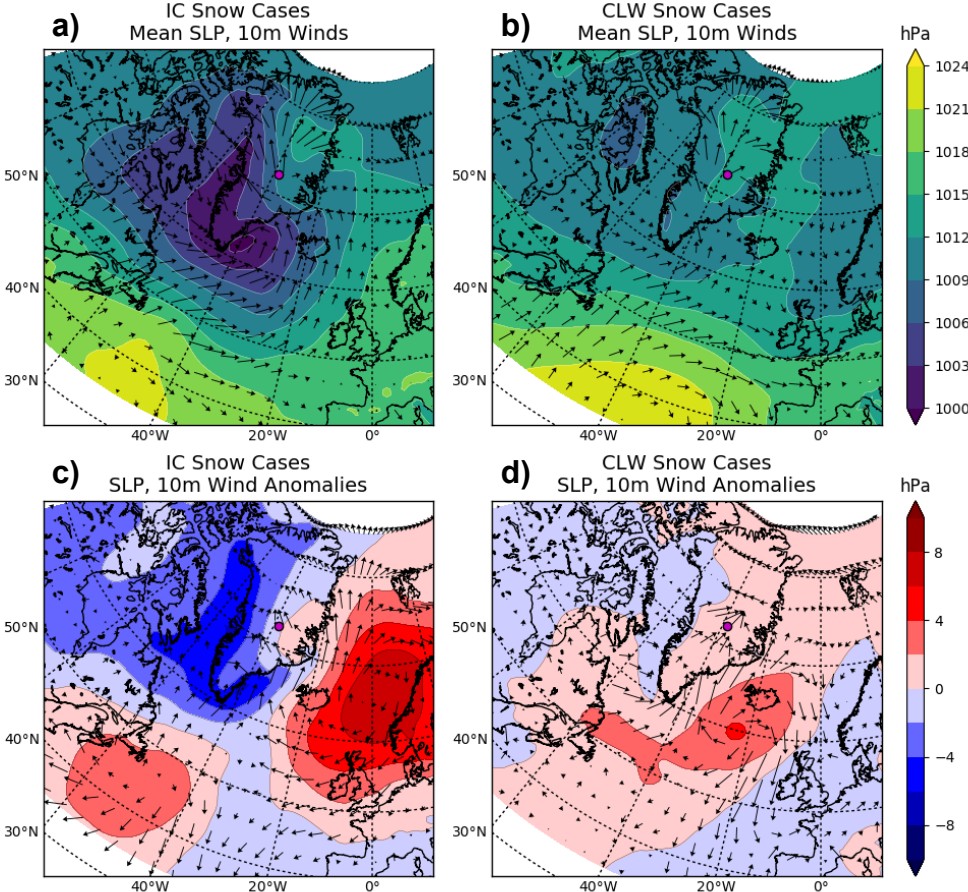

**Figure 10. Panel a (top, left) shows the ERA-Interim derived average SLP and 10 meter winds for 90 IC snow events. Panel b (top, right) shows the same, but for 84 CLW snow events. Both plots are on the same scale. Panels c and d show the anomalies for the SLP and 10 meter winds for the respective cases. The persistent low pressure and strong 10-meter winds are evident for the IC snow cases. In the cases for the CLW snow, there are relatively calm winds and uniform mean SLPs. Both the cyclone and anti-cyclone structure features in the IC snow cases are quite anomalous, whereas the broad high-pressure field in the CLW cases is weakly anomalous.**



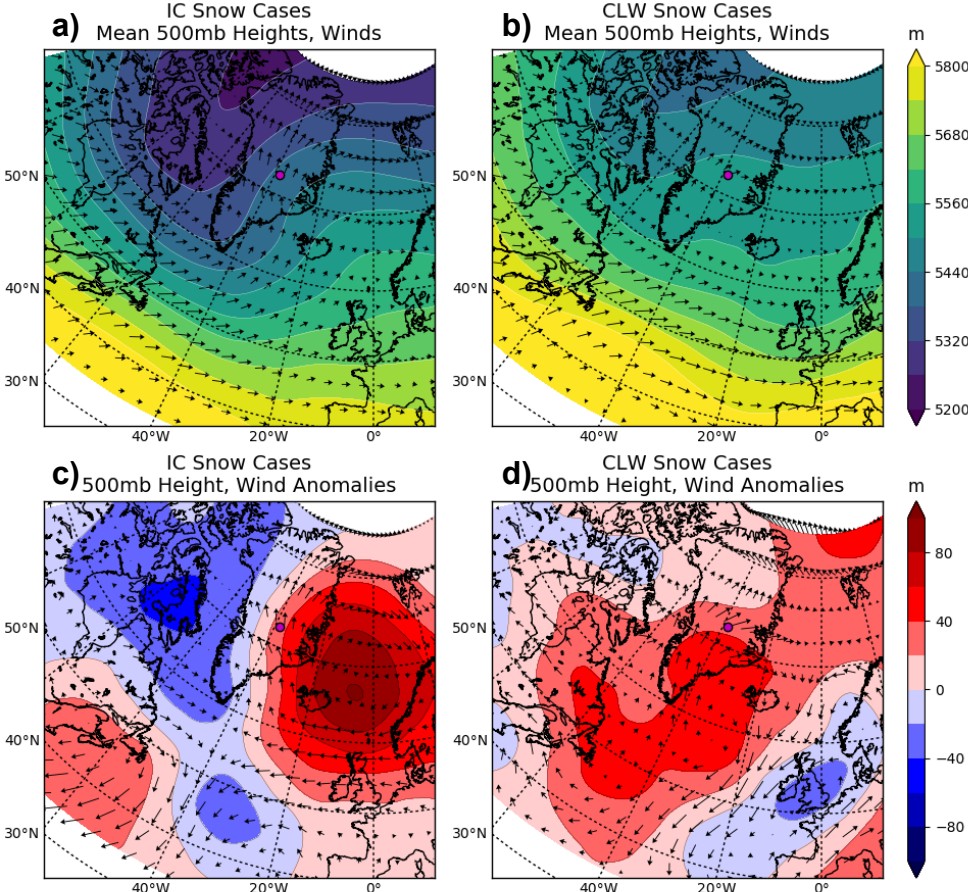

**Figure 11.** Panel a (top, left) shows the ERA-Interim derived average 500mb geopotential heights and winds for 90 IC snow events. Panel b (top, right) shows the same, but for 84 CLW snow events. Both plots are on the same scale. Panels c and d show the anomalies for the 500mb heights and winds for the respective cases. There is an incredibly strong trough and ridge feature in the IC snow cases. This feature indicates diverging upper-level winds just to the east of the trough, over the SE Greenland coast, which would induce strong vertical motions in the column and the upper level winds up show strong SE flow over the GIS. The CLW cases depict relatively calm and flat features, indicating quiescent flow of air up over the GIS from the S and SW.



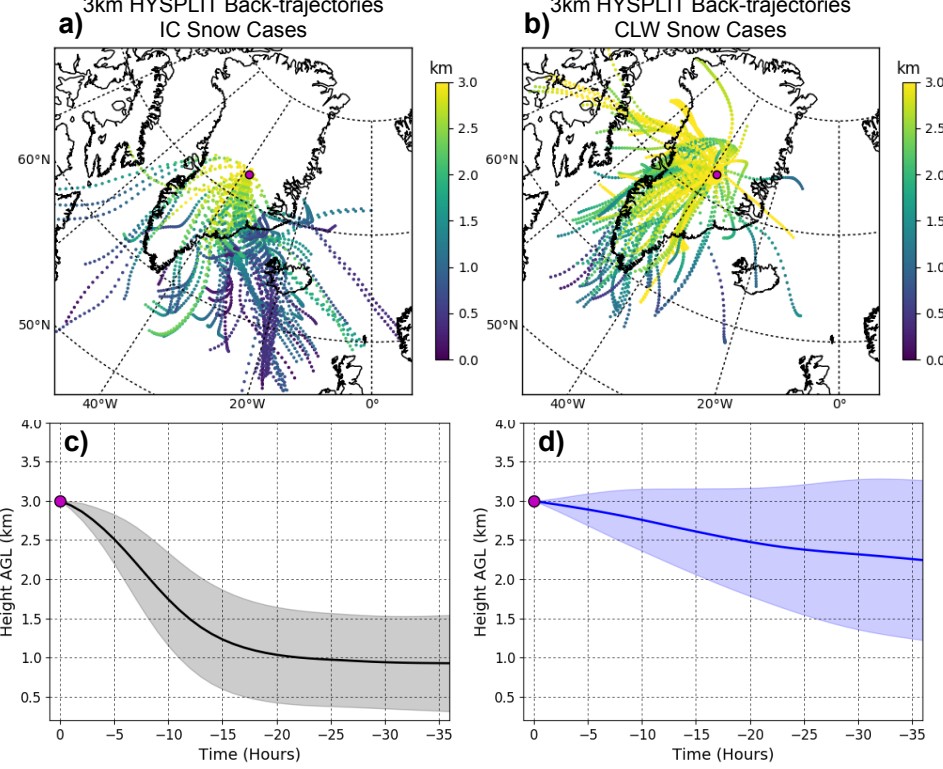

**Figure 12. Panel a (top, left) shows the HYSPLIT calculated, 36 hour backtrajectories for the air at 3 km AGL originating at Summit using GFS for the IC snow cases. The backtrajectory AGL values represent the altitude above ground along each trajectory path. Panel b (top, right) shows the same, but for the CLW cases. The bottom two Panels (c and d), show the mean vertical motions (dark line) and standard deviation (lighter fill) for the IC and CLW cases, respectively. These are consistent with the previous figures: the IC snow cases being vertically lifted and advected over the GIS from the SE. While the CLW cases come from the S and SW along the mean flow.**