# Peer review of "Precipitation regimes over central Greenland inferred from 5 years of ICECAPS observations"

_Atmospheric Chemistry and Physics, 2017_

## Referee Comment (RC1) · Anonymous Referee #1 · 7 Jan 2018

This is a nice and unique analysis of precipitation at Summit, Greenland based on state-of-the-art surface-based passive microwave and cloud radar observations. The clear split of precipitation into two classes is novel, ice dominated processes from the southeast, and mixed phase clouds from the southwest. Distinct circulation features are found with each class. The results parallel some earlier findings by Chen et al. (1997, J. Climate, Precipitation over Greenland Retrieved by a Dynamic Method and Its Relation to Cyclonic Activity). The results present a consistent story. My comments for improvement are very modest.

Specific Comments:

1, Line 5, page 2: Shepherd et al. A Reconciled Estimate of Ice-Sheet Mass Balance. Science, 338, 1183-1189, doi: 10.1126/science.1228102 is a better reference

for Greenland mass balance status.

2. Line 12, page 8: Make clear you are using 1979-2016.

2. Line 8, page 23: Lifted over 5 km on average?

4. The references are hard to read and better formatting is needed.

---

## Referee Comment (RC2) · Anonymous Referee #2 · 2 Feb 2018

This is a well-written, innovate paper. I have only minor questions:

At page 3 line 14 you describe the IC clouds as Ns. I am highly doubtful that the ice precipitation in Ns forms through entirely ice-cloud processes as stated. Ns is associated with heavy precip, and I have yet to see Ns ice precip where liquid somewhere in the profile is not the origin of the 'heavy' precip. For my education, since I'm not an expert on MWR, what is the effect of a deep layer of ice below a thin liquid layer at the top on the measurements (Fig. 1b)? The description in the paper does not address this scenario. Does this produce your indeterminable snow category? Maybe all you need to do is remove the Ns from page 3.

Page 6 L 10: no aggregation in these clouds? I doubt it, see my comments on Figure 6.

[Figure]

Page 7 line 14: define BT

Figure 4: Why is panel c only addressed in a parenthetical comment in the caption?

Page 13 Line 14/14: What does 'per event' mean in this context? These are aggregate results over 5 years?

Page 15 line 20: Narrow is in the eye of the beholder. Please quantify.

Figure 6: What single ice crystal type has, on average, mean fall speeds 1+ m/s as suggested for both IC and CLW clouds? Remember, in the IC clouds you only have vapor deposition and aggregation as available growth processes. I suggest collection growth is far more important in both clouds types than suggested in this paper.

Figure 8: Please discuss the potential for a sampling bias towards light precip events? I suspect it is difficult to get 'good' samples in heavy precip events?

Page 20 line 23: extremely?
* * *

---

## Author Comment (AC1) · 26 Feb 2018

Review of manuscript: "acp-2017-857"

This is a nice and unique analysis of precipitation at Summit, Greenland based on state-of-the-art surface-based passive microwave and cloud radar observations. The clear split of precipitation into two classes is novel, ice dominated processes from the southeast, and mixed phase clouds from the southwest. Distinct circulation features are found with each class. The results parallel some earlier findings by Chen et al. (1997, J. Climate, Precipitation over Greenland Retrieved by a Dynamic Method and Its Relation to Cyclonic Activity). The results present a consistent story. My comments for improvement are very modest.

Pettersen *et al.*: Thank you for the time spent on your thoughtful review and questions and comments. We agree that these results tie in nicely with Chen et al. (1997). We discovered this work after submitting the ACP paper, but we have since used it for our follow-on study. We will add this citation to the discussion, as it is very relevant to the IC snow findings. We will address your comments below (R# is the reply to the comment and M# is the changes made to manuscript if applicable).

M0) Added Chen et al., 1997 citation when referencing Greenland Lee cyclones (Page 21, Line 2 and Page 24, Line 13) and in the reference list, Page 30, Lines 18-19.

1) Line 5, page 2: Shepherd et al. A Reconciled Estimate of Ice-Sheet Mass Balance. Science, 338, 1183-1189, doi: 10.1126/science.1228102 is a better reference for Greenland mass balance status.

R1) Thank you for this updated reference. We will change the citation

M1) Replaced "Tedesco et al., 2011" with "Shepherd et al., 2012" (Page 2, Line 5). Added Shepherd et al., 2012 to references (Page 33, Lines 24-26). Removed Tedesco et al., 2011 from references (Page 34, Line 16).

2) Line 12, page 8: Make clear you are using 1979-2016.

R2) Agreed. We will add the date range

M2) Added (1979 to 2016) to the description (Page 8, Line 12).

3) Line 8, page 23: Lifted over 5 km on average?

R3) This is correct. We set up the HYSPLIT backtrajectory analysis to report the height of each point relative to the ground at that point. So, ground level will change as air moves from over the ocean (0 meters ASL) to Summit Station, Greenland (3200 meters ASL). Since most of the trajectories for the IC snow originate over the ocean at ~1 km ASL (at hour -36 of the backtrajectory – see Figure 12, panels a and

c) and end at 3km above Summit Station (which is >3km ASL), the total delta of lift of the mean parcel (dark line, Figure 12, panel c) is ~5km from hour -36 to hour 0.

M3) We modified the text to better explain the vertical path of the air parcel motion for the IC clouds. We edited the parenthetical comment to say "…(from a mean of 1 km above sea level over the ocean surface, to 3 km AGL over Summit Station, which is approximately 6 km above sea level)…" (Page 23, Lines 8-9).

4) The references are hard to read and better formatting is needed.

R4) We agree that the formatting makes the references hard to distinguish from each other. We used the ACPD template for the paper. We believe that the references will be more clearly separated once the paper has undergone final copy-editing. We will make sure to check the final version for readable references.

**Anonymous Referee #2**

Review of manuscript: "acp-2017-857"

This is a well-written, innovate paper. I have only minor questions:

Pettersen *et al.*: Thank you for the time spent on your thoughtful review and questions and comments. We will attempt to address your points below (R# is the reply to the comment and M# is the changes made to manuscript if applicable).

1) At page 3 line 14 you describe the IC clouds as Ns. I am highly doubtful that the ice precipitation in Ns forms through entirely ice-cloud processes as stated. Ns is associated with heavy precip, and I have yet to see Ns ice precip where liquid somewhere in the profile is not the origin of the 'heavy' precip. For my education, since I'm not an expert on MWR, what is the effect of a deep layer of ice below a thin liquid layer at the top on the measurements (Fig. 1b)? The description in the paper does not address this scenario. Does this produce your indeterminable snow category? Maybe all you need to do is remove the Ns from page 3.

R1) There are two good points in this comment and we will address them separately, below:

1. The description of on Page 3, Line 14 is not well communicated, as we should not be using the word "forms" here. We agree that likely these Ns form with liquid somewhere in the column and associated heavy precipitation is on the SE coastal mountains of Greenland. For those (formerly) Ns clouds/precipitation that make it up and on the GIS and to Summit, we believe these to be fully-glaciated as observed by our instruments (having an immeasurable amount of CLW). We will modify this statement in the description to more clearly show that we are describing the clouds associated with the precipitation as seen at Summit, and not how these systems are formed initially – as that is not possible to say in this analysis.

We will amend the paper to make clear that we are referring to the cloud in the state observed at Summit (not formed) from which the snow is precipitating.

2. At Summit, "heavy" precipitation is still quite light. From Pettersen et al., (2016), we showed that a high Z-Path (column-integrated reflectivity, analogous to IWP) at Summit, is about $10^5$ mm$^6$/m$^2$. Using the same forward model setup in Pettersen et al. 2016, we calculated the BT that would be observed by the MWR with a liquid cloud with and without an ice cloud with $10^5$ Z-Path underneath. At 150 GHz, the increase in BT from the liquid cloud layer is similar (within 15%) with and without the ice layer below. This is due to the relatively low optical depths (less than 1 at 150 GHz) of the atmospheric components (including ice and CLW) in the microwave spectra (see Figure 1, Panel a). We are able to use this unique environment to sort the IC from the CLW originating snowfall regardless of the location (or lack thereof) of the CLW layer (within the measurable limits of the method – see "indeterminate" snow description).

M1) Modified the sentence to indicate that the IC snow is observed to originate from deep, fully-glaciated clouds and removed any mention of formation of the Ns, as we are not observing the formation of these clouds, only the remnant clouds and precipitation that makes it to Summit Station (Page 3, Lines 13-14).

2) Page 6 L 10: no aggregation in these clouds? I doubt it, see my comments on Figure 6.

R2) We agree that no aggregation is too strong of a statement and we do not have enough evidence from the ICEPic to claim this. We believe that the precipitation observed at Summit Station has lower than average aggregation, implies the Matrosov, 2007 Z to S relationship is a good estimate for the MMCR snow rates at this site. See more in R7.

M2) Replaced "non-aggregated" with "lower than average amounts of aggregated…" (Page 6, Line 9).

3) Page 7 line 14: define BT

R3) We originally defined BT on Page 4, Line 24, but it is not mentioned again until Page 7, Line 14 (and it is then reference often). We feel it is better to define BT on Page 7, Line 14 as suggested.

M3) We deleted BT from Page 4, Line 24. We added "brightness temperature (BT)" on Page 7, Line 14.

4) Figure 4: Why is panel c only addressed in a parenthetical comment in the caption?

R4) This was an error – thank you for your careful reading and pointing this out.

We have modified the caption to explain panel c.

M4) Added text to describe Panel c of Figure 4 (Page 41, Lines 5-7).

5) Page 13 Line 14/14: What does 'per event' mean in this context? These are aggregate results over 5 years?

R5) Correct – these are composited results over 5 years.  This is poorly communicated and the use of "event" is ambiguous.  We are trying to communicate that when looking at accumulation statistics in terms of overall %, the IC increases compared to the occurrence % (indicating that it has a relatively higher snow rate) and the indeterminate decreases (occurring often, but with a lower snow rate compared to the other types).  We will remove "event" and try to better communicate this point.

M5) We modified the language describing the snow accumulation (Page 13, Lines 12-13).

6) Page 15 line 20: Narrow is in the eye of the beholder. Please quantify.

R6) We agree that this is ambiguous.  The IC MMCR profiles are narrow when compared to the profiles for the CLW snow.  We will modify the text to add this point and add values for the Doppler Velocities range.

M6) Added in language comparing the IC snow MMCR profiles in comparison to the CLW snow profiles (Page 15, Line 18) and added value for the Doppler Velocities (Page 15, Line 20).

7) Figure 6: What single ice crystal type has, on average, mean fall speeds 1+ m/s as suggested for both IC and CLW clouds? Remember, in the IC clouds you only have vapor deposition and aggregation as available growth processes. I suggest collection growth is far more important in both clouds types than suggested in this paper.

R7) The fact that the average fall speed is +1 m/s and the implications on particle size/collection growth is a very good and important point that we had not before considered.  A few comments related to this observation:

1. Summit Station is located at ~3200 meters ASL and has surface pressure ranging from 620 to 695hPa.  Since the pressure is much lower at Summit Station, equivalent ice particles will fall faster.  Fall speed is proportional to the inverse of the air pressure (Cornford, 1964; Schmidt and Heymsfield, 2009) and therefore particles at Summit Station will have between 1.2 – 1.3 higher fall speeds compared to if they were at sea level Therefore, a fall speed of about 1 m/s at Summit would correspond to a fall speed is closer to a 0.75 – 0.82 m/s fall speed at sea level.
2. Since these fall speeds are obtained from a radar (MMCR), the velocities will be weighted disproportionally towards larger particles.  So, even if there

were only a few aggregates and many non-aggregates, the larger effective diameter particles will bias the mean Doppler velocity values higher.

3. The ice habits observed with ICEPic during the IC events are often bullet rosettes and bullets, which are higher density.

4. We have not seen much evidence of frequent aggregation, but unfortunately we do not have enough data to quantify aggregation occurrence rate. Preliminary data from a few months of Multi-Angle Snowflake Camera (MASC) observations shows very little aggregation and limited riming. And there are too few and too randomly sampled ICEPic images to try and quantify the amount of aggregation. We believe that we cannot conclude the amount of aggregation or lack thereof.

In general, we cannot conclude strongly either way. We do not have enough in-situ evidence to say that there is no aggregation. However, we do see lack of evidence of frequent aggregation in the ICEPic and MASC images.

8) Figure 8: Please discuss the potential for a sampling bias towards light precip events? I suspect it is difficult to get 'good' samples in heavy precip events?

R8) We agree that there are likely sampling biases in Figure 8. However, we feel that the bias is likely towards heavier (obvious) snowfall during lower-wind conditions at Summit Station. Heavy events at Summit Station would be classified as very light snow in the mid-latitudes (a classification of IC+ in NWS METAR terminology). Meanwhile, light precipitation events at Summit are so light that one may not even notice that there is snowfall occurring. However, during the windier snow events the technician may be unable to gather samples – the gathering of ice crystals on a slide would be too difficult (and possibly unsafe due to limited visibility) and there would be contamination on the slides from blowing snow. We think that it is appropriate to add that the process of gathering ice crystal pictures is biased towards events with lighter winds and may decrease the sampling in some of the windier storms, which could correspond to some of the heavier events (though this point was not explored in this study).

M8) Added to the caption information that the IcePIC images are biased to lower-wind conditions (Page 45, Lines 2-3)

9) Page 20 line 23: extremely?

R9) Agreed – the use of extremely is too strong here.

M9) Deleted "extremely" (Page 20, Line 18)